# Evocalcet prevents ectopic calcification and parathyroid hyperplasia in rats with secondary hyperparathyroidism

**Mariko Sakai[1], Shin Tokunaga[1], Mika Kawai[2], Miki Murai[2], Misaki Kobayashi[2], Tetsuya Kitayama[1], Satoshi Saeki[1], Takehisa Kawata**[1] *

**1** Nephrology Research Laboratories, Nephrology R&D Unit, R&D Division, Kyowa Kirin Co., Ltd., Shizuoka, Japan, **2** Research Core Function Laboratories, Research Functions Unit, R&D Division, Kyowa Kirin Co., Ltd., Shizuoka, Japan

* takehisa.kawata.kk@kyowakirin.com

**Data Availability Statement:** All relevant data are within the paper and its Supporting Information files.

## Abstract

### Background

Elevated parathyroid hormone (PTH) levels in secondary hyperparathyroidism (SHPT) lead to vascular calcification, which is associated with cardiovascular events and mortality. Increased PTH production is caused by the excessive proliferation of parathyroid gland cells, which is accelerated by abnormal mineral homeostasis. Evocalcet, an oral calcimimetic agent, inhibits the secretion of PTH from parathyroid gland cells and has been used for the management of SHPT in dialysis patients. We observed the effects of evocalcet on ectopic calcification and parathyroid hyperplasia using chronic kidney disease (CKD) rats with SHPT.

### Methods

CKD rats with SHPT induced by adenine received evocalcet orally for 5 weeks. The calcium and inorganic phosphorus content in the aorta, heart and kidney was measured. Ectopic calcified tissues were also assessed histologically. To observe the effects on the proliferation of parathyroid gland cells, parathyroid glands were histologically assessed in CKD rats with SHPT induced by 5/6 nephrectomy (Nx) after receiving evocalcet orally for 4 weeks.

### Results

Evocalcet prevented the increase in calcium and inorganic phosphorus content in the ectopic tissues and suppressed calcification of the aorta, heart and kidney in CKD rats with SHPT by reducing the serum PTH and calcium levels. Evocalcet suppressed the parathyroid gland cell proliferation and reduced the sizes of parathyroid cells in CKD rats with SHPT.

### Conclusions

These findings suggest that evocalcet would prevent ectopic calcification and suppress parathyroid hyperplasia in patients with SHPT.

**Funding:** Mitsubishi Tanabe Pharma Corporation provided evocalcet. All studies were performed and the cost of them were supported by Kyowa Kirin Co., Ltd. Mariko Sakai, Shin Tokunaga, Mika Kawai, Miki Murai, Misaki Kobayashi, Tetsuya Kitayama, Satoshi Saeki, and Takehisa Kawata are employees of Kyowa Kirin Co., Ltd. Kyowa Kirin Co., Ltd provided support in the form of salaries for authors, MS, ST, MK, MM, MK, TK, SS and TK, but did not have any additional role in the study design, data collection and analysis, decision to publish, or preparation of the manuscript. The specific roles of these authors are articulated in the 'author contributions' section.

**Competing interests:** Evocalcet is the product in development, and Kyowa Kirin Co., Ltd and Mitsubishi Tanabe Pharma Corporation have the ownership of the patents of evocalcet. This does not alter our adherence to PLOS ONE policies on sharing data and materials.

## Introduction

Secondary hyperparathyroidism (SHPT), which is characterized by serum parathyroid hormone (PTH) elevation, is a common mineral metabolism abnormality in patients with chronic kidney disease (CKD). Excessive PTH secretion disturbs the calcium (Ca) and phosphate metabolism, which is considered to be the main cause of ectopic calcification [1]. Accelerated coronary artery calcification is often found in dialysis patients with SHPT and its presence is associated with increased cardiovascular events and mortality [2]. Since there is no treatment that can efficiently reverse vascular calcification, its prevention is important in improving prognosis of patients with SHPT.

Parathyroid hyperplasia is also a characteristic feature of SHPT and its progression leads to hypersecretion of PTH from the parathyroid glands and this in turn leads to altered mineral metabolism [3]. Hence, preventing the progression of parathyroid cell growth and gland enlargement is important to suppress the elevation of serum PTH and Ca.

Cinacalcet is the first approved calcimimetic agent in 2004, allosterically modulates the calcium receptor (CaR) on parathyroid gland cells and suppresses PTH secretion [4, 5]. Cinacalcet has been widely used clinically for more than 10 years in the world for management of SHPT in dialysis patients [6–11] and succeeded to reduce the number of parathyroidectomies [12]. It has been reported that the progression of cardiovascular calcification was attenuated by combination therapy with cinacalcet and low-dose vitamin D receptor activator (VDRA) in comparison to monotherapy with higher doses of VDRA in SHPT patients [13]. It has also been suggested that cinacalcet contributes to reducing the size of the parathyroid glands in SHPT patients [14–16].

Although cinacalcet has excellent PTH lowering effect, it has problems of causing upper gastro-intestinal (GI) side effects such as nausea and vomiting at a certain rate [17]. These problems sometimes become an obstacle to the long-term use of cinacalcet or treatment with increased doses of cinacalcet [18, 19]. Evocalcet has been developed to address these issues. Although evocalcet suppresses PTH secretion with a similar pharmacological profile to cinacalcet, it has lesser effect on the GI tract than cinacalcet [20]. Nevertheless, the effects of evocalcet on ectopic calcification and parathyroid hyperplasia need to be evaluated further.

In this study, we assessed the effects of evocalcet on vascular and tissue calcification using CKD rats with SHPT induced by adenine feeding. We also measured the effects of evocalcet on parathyroid gland cell proliferation and cell size in CKD rats with SHPT induced by 5/6 nephrectomy (Nx).

## Materials and methods

### Evocalcet

Evocalcet was synthesized at Mitsubishi Tanabe Pharma Corporation (Lot No. 134016 for the single administration study, Lot No. 45AP80110001 for the long term administration studies, Osaka, Japan).

### Animals

Male Sprague Dawley rats (13 weeks of age) were purchased from Japan SLC Inc. (Shizuoka, Japan) for the study using CKD rats with SHPT induced by adenine. Male Sprague Dawley rats (6 weeks of age) were purchased from Charles River Laboratories Japan, Inc. (Kanagawa, Japan), for the study using CKD rats with SHPT induced by 5/6 Nx.

The rats were kept at 20–26°C and 30–70% humidity under a 12-hour light-dark cycle with *ad libitum* access to tap water and commercial chow (for the study using CKD rats with SHPT

induced by adenine: CE-2, CLEA Japan, Inc., Shizuoka, Japan. For the study using CKD rats with SHPT induced by 5/6 Nx: FR-2, Funabashi Farm Co., Ltd., Chiba, Japan. Both diets are normal diets) prior to the experiments. All experimental animals were monitored at least once per working day throughout the course of the study. Humane endpoints were defined as reduced physical activity level, weight loss, hunched posture, and other signs of distress. All rats reaching humane endpoints or in the single administration study were euthanized by carbon dioxide inhalation after the completion of studies. Euthanasia by carbon dioxide inhalation was conducted in the home cage. An optimal flow rate is 20% replacement of the home cage volume/min. We observed the respiratory and cardiac arrest in rats, and maintained $CO_2$ flow for at least 3 minutes after respiratory and cardiac arrest. After both signs were observed, rats were removed from the cage. The rats in the long term studies were euthanized by exsanguination via the abdominal aorta/vena cava under isoflurane anaesthesia. All animal studies were carried out in strict accordance with the Standards for Proper Conduct of Animal Experiments at Kyowa Kirin Co., Ltd. The protocol was approved by the Institutional Animal Care and Use Committee (IACUC) of Kyowa Kirin Co., Ltd. (protocol number APS 18J0188 for the single administration study, 17J0078 for the five-week administration study using CKD rats with SHPT induced by adenine, 14J0052 for the four-week administration study using CKD rats with SHPT induced by 5/6 Nx), and all efforts were made to minimize patient distress and suffering.

## CKD rats with SHPT induced by adenine

**Single administration study.** To establish CKD rats with SHPT induced by adenine, eighteen rats were fed with a CE-2 diet containing 0.75% adenine and 2.5% protein (adenine diet; CLEA, Japan, Inc., Shizuoka, Japan). Six rats in the control group were fed with a CE-2 diet containing 25% protein (control diet). After three weeks of the adenine-diet feeding, rats were randomly divided into three groups matched for body weight as well as blood urea nitrogen (BUN) and serum creatinine. The adenine diet was then changed to a normal diet and vehicle (0.5% methyl cellulose solution) or evocalcet (0.03 or 0.3 mg/kg) was orally administered. Blood samples were obtained from the tail vein before and 2, 4, 8, and 24 hours after the administration.

**Five-week administration study.** CKD rats with SHPT induced by adenine by the methods described above, were used. After adenine-diet feeding, sixteen rats were randomly divided into two groups. The adenine diet was then changed to a normal diet, and vehicle (0.5% methyl cellulose solution) or evocalcet (0.3 mg/kg) were orally administered once daily for five weeks. Blood samples were obtained from the jugular vein 24 hours after the last administration. At the end of the study, the thoracic aorta, abdominal aorta, heart and kidney were removed and their Ca and inorganic phosphorus (IP) content and calcification levels were measured.

**Biochemical analyses.** The serum PTH levels were measured using a Rat Intact PTH ELISA kit (Immutopics, Inc., San Clemente, CA). The serum Ca, IP, BUN and creatinine levels were measured using an auto analyzer (Hitachi High-Technologies Corporation., Tokyo, Japan). For the single administration study, the serum Ca level was measured using a Calcium E-test Wako (FUJIFILM Wako Pure Chemical Co., Ltd., Osaka, Japan).

**Evaluation of the Ca and IP content in the thoracic aorta, heart and kidney.** The thoracic aorta, heart and kidney were defatted with chloroform and methanol (2:1) for two days and dehydrated by acetone for three hours. The samples were incinerated to ashes at 550˚C for 12 hours using an electric muffle furnace, then extracted with hydrochloric acid and diluted with distilled water. The levels of Ca and IP in the tissue were measured using a Calcium E-test

Wako and Phospha C-test Wako (FUJIFILM Wako Pure Chemical Co., Ltd., Osaka, Japan) respectively and were represented as the weight of Ca or IP per dry tissue weight.

**Evaluation of calcification with von Kossa staining.** The thoracic aorta, abdominal aorta, heart and kidney were fixed in a 10% neutral-buffered formalin and embedded in paraffin and sectioned by standard methods. Paraffin blocks were sectioned into slices of approximately 3 μm in thickness. The sections were stained using the von Kossa method and scored by visually estimating the percentage of the stained area within the samples as: 0% (none), ±: <25% (slight), +: 25–50% (mild), 2+: 50–75% (moderate), 3+: >75% (marked).

## CKD rats with SHPT induced by 5/6 Nx

**Four-week administration study.** Rats were 5/6 nephrectomized in two steps. Under anesthesia (pentobarbital, 50 mg/kg; intraperitoneally) and analgesia (lidocaine; topically), two-thirds of the left kidney was removed, and then the right kidney was removed after a seven-day interval. Seven days after the completion of 5/6 Nx, the FR-2 diet was changed to a high-phosphate diet, containing 0.6% calcium and 0.9% phosphate (Oriental Yeast Co., Ltd., Tokyo, Japan). Approximately two weeks after the initiation of the high-phosphate diet, sixty four 5/6 Nx rats were divided into three groups matched for body weight as well as BUN and serum PTH and Ca. Vehicle or evocalcet (0.1 or 0.3 mg/kg) was orally administered to each group once daily for four weeks. To evaluate the cellular proliferation, the rats were subcutaneously infused with 5-bromo-2'-deoxyuridine (BrdU) by osmotic pump from seven days before necropsy. At necropsy, the weight of the parathyroid gland was measured.

**Evaluation of parathyroid gland cell proliferation and cell size.** The parathyroid glands were fixed with 10% (vol) neutral-buffered formalin and routinely processed with paraffin. The paraffin blocks were sectioned into slices of approximately 5 μm in thickness. After dewaxing and rehydration, the sections were incubated with 3% (vol) hydrogen peroxide/PBS, to quench the endogenous peroxidase, and treated with 0.05% (w/v) pronase E for deproteinization. The sections were incubated with an anti-BrdU antibody (Clone: Bu20a; DAKO Inc., Carpinteria, CA) and with horseradish peroxidase-labelled polymer conjugated to anti-mouse immunoglobulins (Envision+; DAKO Inc.). The signal was visualized with 3,3-diaminobenzidine. Hematoxylin was applied for nuclear counterstaining.

The ratio of BrdU-positive cells was calculated from the number of BrdU-positive nuclei, the number of total nuclei, and the area of a section. The area of parathyroid gland cells was expressed as the cell size and determined using a hematoxylin-stained section. The cell size was calculated using the following formula: Cell size ($\mu m^2$) = the total area of section/the total number of nucleus in a section. These parameters were analyzed using the Aperio ImageScope and Nuclear (ver. 9) software programs (Aperio Technologies Inc., Vista, CA).

## Statistical analysis

All values are expressed as the mean + S.E. The statistical analyses were all performed using a statistical analysis software program (SAS, release 9.4; SAS Institute, Inc., Cary, USA). Differences between the control or sham groups and the vehicle-treated CKD rat group were determined by Fisher's $t$ test followed by Student's $t$-test or the Aspin-Welch test. When significant differences between the vehicle-treated CKD rat group and the evocalcet-treated groups were identified by Bartlett test, differences were determined by a one-way ANOVA followed by Dunnett's test. When significant differences were not identified by Bartlett test, differences were assessed by the Kruskal-Wallis test followed by the Steel test. P values of <0.05 were considered to indicate statistical significance in all of the analyses.

## Results

### Effects of evocalcet on serum PTH and Ca in CKD rats with SHPT induced by adenine (single administration study)

We observed the effects of evocalcet on serum PTH and Ca in CKD rats with SHPT induced by adenine. The rats, which were fed an adenine diet for three weeks, showed a significant increase in serum PTH levels in comparison to control rats (7750±901 pg/mL vs. 2737±592 pg/mL), suggesting the development of SHPT. Oral treatment with evocalcet (0.3 mg/kg) obviously decreased the serum PTH levels in comparison to vehicle-treated CKD rats at 2 hours after administration; the effect lasted for 24 hours (Fig 1A). The serum Ca levels were also decreased by treatment with evocalcet (0.3 mg/kg) (Fig 1B). On the other hand, evocalcet (0.03 mg/kg) did not clearly affect either serum PTH or Ca at any point (Fig 1A and 1B).

### Effects of evocalcet on biochemical parameters in CKD rats with SHPT induced by adenine (five-week administration study)

CKD rats with SHPT induced by adenine were treated with evocalcet for five weeks. There were no differences in body weight or food consumption between vehicle-treated CKD rats and evocalcet-treated CKD rats (Table 1).

At the end of the study period, in comparison to the control rats, CKD rats exhibited significant increases in their serum PTH and IP levels as well as BUN and serum creatinine levels (Table 1). Evocalcet treatment reduced the serum PTH and Ca levels, while the serum IP level tended to increase, resulting in no significant differences in the serum calcium phosphate product (Ca x IP), which is presumed to be a risk factor for aortic calcification. The relative heart and kidney weight were increased in vehicle-treated CKD rats; no significant improvement was observed in the evocalcet-treated group.

### Effects of evocalcet on ectopic calcification in CKD rats with SHPT induced by adenine

In order to examine the effectiveness of evocalcet in preventing ectopic calcification of soft tissues, we analyzed the Ca and IP levels of the thoracic aorta, heart and kidney, and calcification of the thoracic aorta, abdominal aorta, heart and kidney histologically by von Kossa staining. In the vehicle-treated CKD rats group, the tissue Ca and IP content in the thoracic aorta was significantly increased in comparison to the control group (Fig 2A and 2B), which displayed Mönckeberg arterial calcification, characterized by Ca deposition in the medial smooth muscle

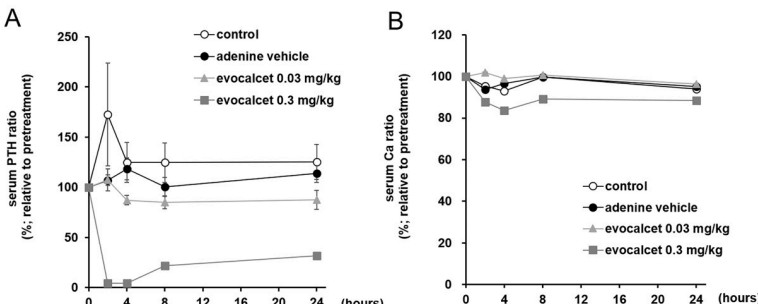

**Fig 1. Effects of evocalcet on serum PTH and Ca in CKD rats (single administration study).** Vehicle or evocalcet (0.03 or 0.3 mg/kg) was orally administered to control rats and CKD rats with SHPT induced by adenine. (A) Serum PTH and (B) Ca levels. The data are presented as the mean + S.E. (n = 6/group).

**Table 1. Biochemical parameters of evocalcet after 5-week-administration to CKD rats.**

| Parameters | Sham | Adenine | |
|---|---|---|---|
| | | Vehicle | Evocalcet |
| body weight (BW; g) | 500 ± 17 | 370 ± 15## | 394 ± 8 |
| cumulative food intake (g) | 1265 ± 33 | 791 ± 28## | 801 ± 11 |
| serum PTH (pg/mL) | 991 ± 235 | 7630 ± 2020# | 1359 ± 416** |
| serum Ca (mg/dL) | 10.7 ± 0.1 | 10.5 ± 0.2 | 10.0 ± 0.1 |
| blood urea nitrogen (mg/dL) | 20.9 ± 0.8 | 109.5 ± 11.7## | 121.1 ± 13.0 |
| serum creatinine (mg/dL) | 0.41 ± 0.01 | 1.84 ± 0.23## | 2.05 ± 0.24 |
| serum IP (mg/dL) | 7.1 ± 0.2 | 9.1 ± 0.8# | 9.7 ± 0.7 |
| Ca x IP (mg2/dL2) | 75.9 ± 2.7 | 94.4 ± 6.7# | 97.5 ± 7.5 |
| heart weight (mg/g BW) | 2.45 ± 0.03 | 3.76 ± 0.23## | 3.58 ± 0.11 |
| kidney weight (mg/g BW) | 5.5 ± 0.2 | 22.5 ± 3.3## | 22.6 ± 2.2 |

[#]$P < 0.05$,

[##]$P < 0.01$ vs. control group (Student's t-test or Aspin-Welch test);

[**]$P < 0.01$ vs. vehicle-treated CKD rat group (Steel test).

cells (Fig 3). Evocalcet reduced the content of both Ca and IP, as well as calcification of the aorta in CKD rats (Figs 2A, 2B and 3), which was also shown by von Kossa staining score (Table 2). Similarly, the tissue Ca content and calcification scores in the heart and kidney were increased in CKD rats in comparison to the control group, and were reduced by the administration of evocalcet (Fig 2C–2F and Table 2).

## Effects of evocalcet on parathyroid gland cell proliferation and size in CKD with SHPT rats induced by 5/6 Nx

Next, to evaluate the effects of evocalcet on parathyroid gland cell proliferation, CKD rats with SHPT induced by 5/6 Nx were treated with evocalcet (0.1 and 0.3 mg/kg) once daily for four weeks. The dose of 0.1 mg/kg was associated with a significant reduction in serum PTH and Ca levels and an even higher reduction was observed at a dose of 0.3 mg/kg in this model, as we showed previously [20]. The ratio of BrdU-positive cells (%) in the vehicle-treated CKD rat group was significantly increased in comparison to the sham group (Figs 4 and 5). Parathyroid gland cell proliferation was suggested to be induced by 5/6 Nx. Evocalcet treatment significantly decreased the ratio of BrdU-positive cells in comparison to vehicle-treated CKD rats. The comparison of the vehicle-treated CKD rat group and the sham group revealed that the weight of the parathyroid gland and the size of parathyroid gland cells were significantly increased, whereas these increases were significantly ameliorated by evocalcet treatment (Fig 5).

## Discussion

Evocalcet is a newly synthesized calcimimetic compound that improves several issues with cinacalcet, including the high rate of GI tract side-effects, the inhibition of the CYP2D6 enzyme and low bioavailability due to rapid metabolization by the CYP3A4 enzyme. Meanwhile, its pharmacological profile as an allosteric modulator of CaR and the effect of suppressing PTH secretion from parathyroid gland cells are similar to cinacalcet [17, 21–24].

In this study, we investigated whether evocalcet would prevent ectopic calcification and suppress parathyroid hyperplasia. We used CKD rats with SHPT induced by adenine to evaluate the effect on calcification because progressive ectopic calcification is induced in this model

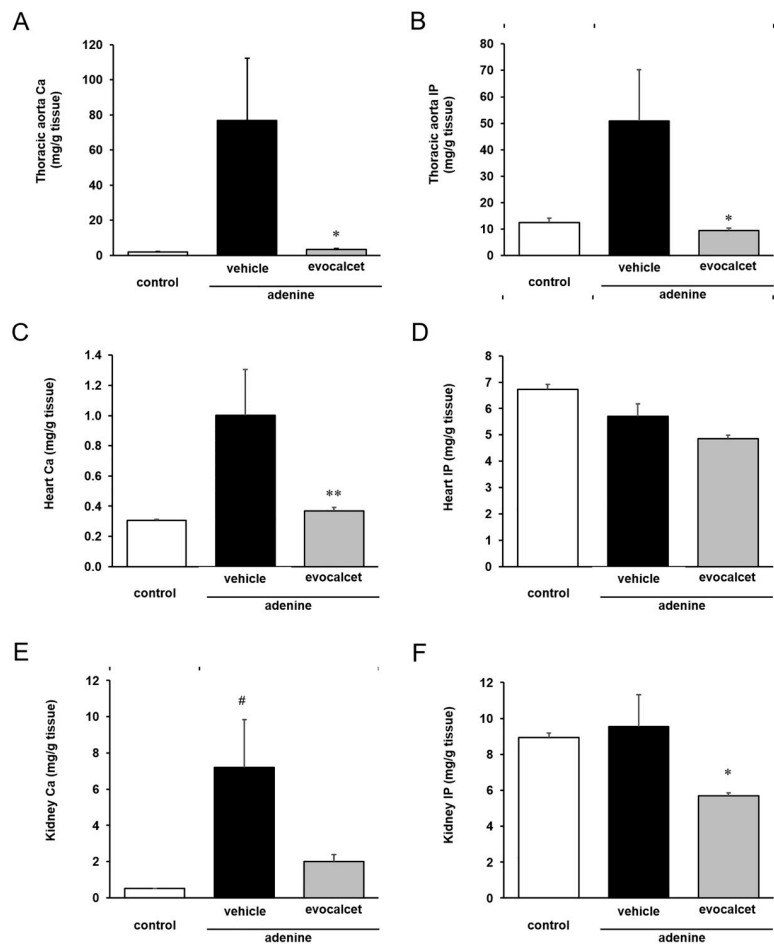

**Fig 2. Effects of evocalcet on Ca and IP content in tissues in CKD rats.** Vehicle or evocalcet (0.3 mg/kg) was orally administered to control and CKD rats with SHPT induced by adenine once daily for five weeks. The data are presented as the mean + S.E. (n = 5-10/group). (A) The Ca and (B) IP content in the thoracic aorta. (C) The Ca and (D) IP content in the heart. (E) The Ca and (F) IP content in kidney. #P < 0.05 vs. control group (Aspin-Welch test); *P < 0.05, and **P < 0.01 vs. vehicle-treated CKD rat group (Steel test).

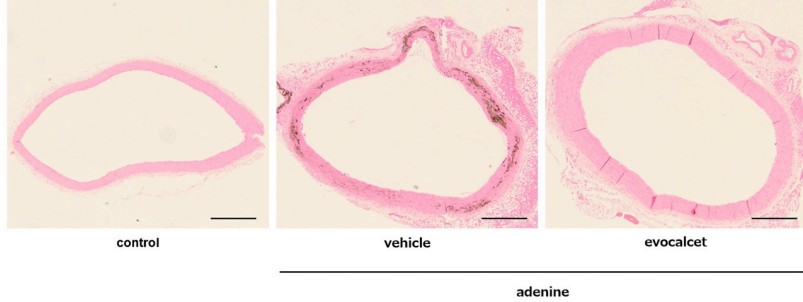

**Fig 3. Representative von Kossa staining of the CKD rat aorta.** Thoracic aortas from control rats and CKD rats with SHPT induced by adenine were subjected to von Kossa staining. Scale bar, 500 μm. (A) Control; (B) vehicle-treated CKD; (C) evocalcet-treated CKD.

**Table 2. Von Kossa stain scoring.**

| | Grade | | | | |
|---|---|---|---|---|---|
| | - | ± | + | 2+ | 3+ |
| Thoracic aorta | | | | | |
| control | 5 | 0 | 0 | 0 | 0 |
| adenine vehicle | 3 | 1 | 1 | 2 | 2 |
| adenine evocalcet | 10 | 0 | 0 | 0 | 0 |
| Abdominal aorta | | | | | |
| control | 5 | 0 | 0 | 0 | 0 |
| adenine vehicle | 4 | 1 | 0 | 2 | 2 |
| adenine evocalcet | 10 | 0 | 0 | 0 | 0 |
| Heart | | | | | |
| control | 5 | 0 | 0 | 0 | 0 |
| adenine vehicle | 6 | 1 | 2 | 0 | 0 |
| adenine evocalcet | 9 | 1 | 0 | 0 | 0 |
| Kidney | | | | | |
| control | 5 | 0 | 0 | 0 | 0 |
| adenine vehicle | 0 | 0 | 4 | 5 | 0 |
| adenine evocalcet | 0 | 2 | 8 | 0 | 0 |

The data are presented as the number of samples with each von Kossa staining score for each group. Scoring: -: 0% (none), ±: <25% (slight), +: 25–50% (mild), 2+: 50–75% (moderate), 3+: >75% (marked).

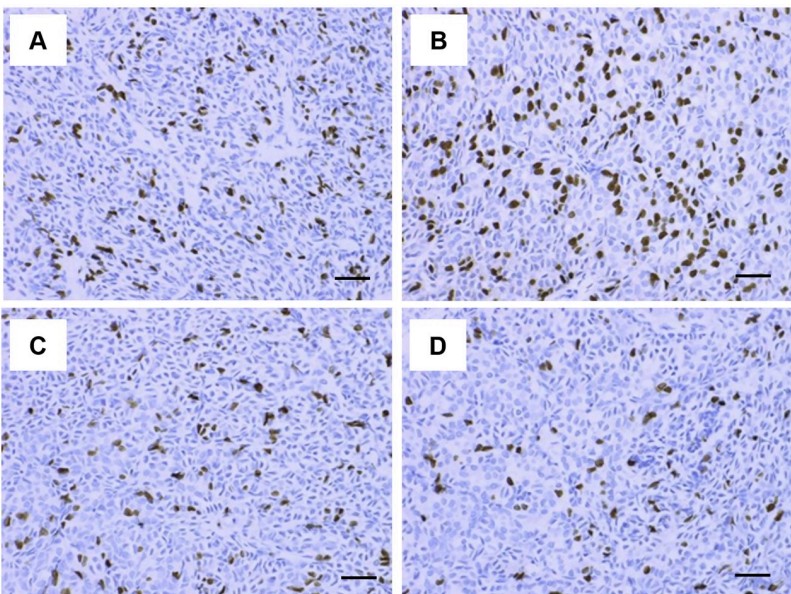

**Fig 4. Representative BrdU staining of a parathyroid gland specimen from a CKD rats.** The parathyroid glands from sham-operated and CKD rats with SHPT induced by 5/6 Nx were stained with BrdU. (A) Sham; (B) vehicle-treated CKD; (C) evocalcet-treated CKD (0.1 mg/kg); (D) evocalcet-treated CKD (0.3 mg/kg). Scale bar, 50 μm. Nx, nephrectomy.

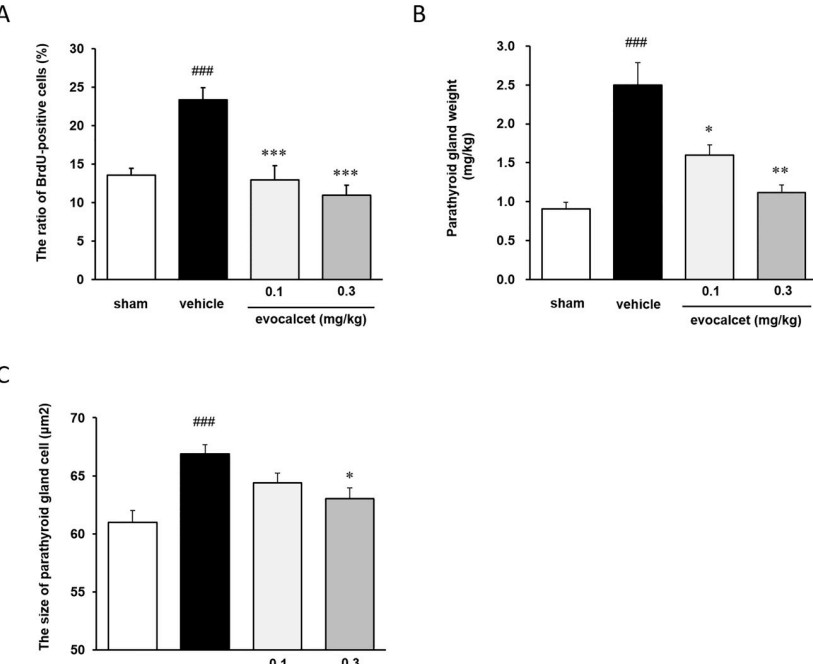

**Fig 5. Effects of evocalcet on parathyroid gland cell proliferation and sizes in CKD rats.** Vehicle or evocalcet (0.1 or 0.3 mg/kg) was orally administered to sham-operated and CKD rats with SHPT induced by 5/6 Nx once daily for four weeks. The data are presented as the mean + S.E. (n = 10-11/group). (A) Parathyroid gland cell proliferation. (B) Parathyroid gland weight. (C) The size of the parathyroid gland cells. ###P < 0.001 vs. Sham group (Student's t-test or Aspin-Welch test); *P < 0.05, **P < 0.01, and ***P < 0.001 vs. vehicle-treated CKD group (Dunnett's test or Steel test). Nx, nephrectomy.

by high levels of serum PTH followed by the accumulation of Ca and phosphate in soft tissues [25]. The dosage of evocalcet (0.3 mg/kg) that showed clear suppressive effect on serum PTH and Ca levels by single administration was used in this study [20]. As a result of five weeks continuous treatment, evocalcet reduced the Ca and IP content and prevented calcification of the aorta, heart and kidney. This preventive effect of evocalcet on calcification is likely caused by the inhibition of PTH secretion from parathyroid gland cells followed by the reduction of serum Ca, since we previously showed that PTH depletion by parathyroidectomy suppressed calcification in CKD rats with SHPT induced by 5/6 Nx [26].

Although evocalcet tends to decrease the serum phosphate levels in patients with SHPT [27], we could not observe the reduction in serum IP levels in CKD rats with SHPT induced by adenine. The discrepancy is possibly due to the remnant kidney function of this animal model of CKD. In fact, due to the negation of phosphaturic effect of PTH by these calcimimetic agents, phosphorus levels actually increase in CKD patients [28]. Since PTH reduces phosphate reabsorption at the proximal renal tubule, the inhibition of PTH by evocalcet might reduce the excretion of phosphate from the renal tubules [29, 30], which negates the decrease in serum phosphate level resulting from the prevention of bone resorption.

Ectopic calcification is formed at an early stage of SHPT, and is accelerated by the increase of serum Ca and phosphate levels and associated with high-turnover bone disease with high serum PTH levels [1, 31, 32]. The processes involved in vascular calcification have been identified [33]. Briefly, Ca is actively deposited in the vessel wall and forms amorphous calcium phosphate under hypercalemia. This compound gradually transforms into the insoluble apatite

form within tissue leading to vascular calcification. In circulation, calcium phosphate crystals coexist with chaperone-binding proteins that inhibit the crystallization of Ca and IP particles at physiological concentrations. These mineral-protein complexes are called calciprotein particles, CPP, and contain fetuin-A and Matrix Gla protein, well-known circulating inhibitors of calcification [34, 35]. Since it was suggested that suppressive effect of evocalcet on serum Ca contributed to prevent the ectopic calcification, there is room for further study to evaluate whether evocalcet suppresses CPP production and there is the effects of evocalcet on these factors for calcification.

We also observed the effect of evocalcet on parathyroid gland growth using CKD rats with SHPT induced by 5/6 Nx. We found that evocalcet significantly suppressed the parathyroid gland enlargement and cell proliferation. By reducing parathyroid hyperplasia, evocalcet might suppress the development of SHPT, thereby inhibiting the increase of serum PTH and thus prevent ectopic calcification at an initial stage.

Reduced expression of the CaR is one of the main cause of proliferation of parathyroid grand cells and cinacalcet shows the preventive effect on the growth of parathyroid gland cells [36–39]. NPS R-568, a calcimimetic compound, was shown to reduce parathyroid gland cell volume in CKD rats with SHPT induced by 5/6 Nx [40]. These reports suggest that activation of CaR is important in the suppression of parathyroid cell proliferation.

Parathyroid hyperplasia is caused by a complex cascade of events which has not been completely clarified. Nevertheless, there are some hypotheses that can explain the regression of the parathyroid gland by calcimimetic agents. First, calcimimetic agents may affect the cell cycle. Cinacalcet has been shown to increase the expression of p21, a cell cycle inhibitor, which controls cell entry into the actively dividing S phase, and thus prevents parathyroid gland cells from entering a proliferative hyperplastic state [37, 39]. Second, calcimimetic agents may promote cell apoptosis. High concentrations of cinacalcet have been shown to induce apoptosis in parathyroid cells from uremic rats in vitro, although this result is controversial [14]. Third, calcimimetic agents may increase the oxyphil/chief cell ratio in the parathyroid gland. Parathyroid glands are composed of many chief cells and fewer oxyphil cells, and the correlation between cinacalcet therapy and a high oxyphil/chief cell ratio was shown in hemodialysis patients. Since the oxyphil cells have a lower level of proliferation than the chief cells, the increase in the oxyphil/chief cell ratio may reduce parathyroid gland cell proliferation [41]. As a calcimimetic agent that allosterically activates the CaR, these hypotheses might be applied to evocalcet. However, the size of the oxyphil cell is larger than that of the chief cell, so none of these hypotheses explains why the parathyroid gland cell sizes were reduced in this study. Further studies are needed to verify the effect of evocalcet on these factors.

In summary, the present study suggests that evocalcet can improve ectopic calcification and parathyroid hyperplasia by inhibiting PTH secretion in SHPT patients similarly to cinacalcet. These effects seemed important to improve prognosis in patients with SHPT. Evocalcet is therefore expected to significantly contribute to the management of SHPT through early intervention.

## Supporting information

**S1 Table. The set of raw data for Fig 1.**
(XLSX)

**S2 Table. The set of raw data for Fig 2.**
(XLSX)

**S3 Table. The set of raw data for Fig 5.**
(XLSX)

**S4 Table. The set of raw data for Table 1.**
(XLSX)

**S5 Table. The set of raw data for Table 2.**
(XLSX)

## Acknowledgments

We thank Mitsubishi Tanabe Pharma Corporation for providing evocalcet. We thank Yasu-hiro Ina, Yuki Tanbo, Miho Araki and Youji Shoukei for their assistance in the in vivo studies. We also thank Toyoko Kashiwagi and Naoya Kimoto for their support in pathological analysis. Ecpotic calcification study was supported by Takahiro Sugiura (Nihon Bioresearch Inc.).

## Author Contributions

**Conceptualization:** Mariko Sakai, Shin Tokunaga, Tetsuya Kitayama, Satoshi Saeki, Takehisa Kawata.

**Data curation:** Mariko Sakai, Shin Tokunaga, Mika Kawai, Miki Murai, Misaki Kobayashi, Takehisa Kawata.

**Methodology:** Takehisa Kawata.

**Writing – original draft:** Mariko Sakai.

**Writing – review & editing:** Shin Tokunaga, Tetsuya Kitayama, Satoshi Saeki, Takehisa Kawata.

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
