## [Decision Letter · Decision Letter 0]

24 Feb 2020

PONE-D-20-01532

Evocalcet prevents ectopic calcification and parathyroid hyperplasia in rats with secondary hyperparathyroidism.

PLOS ONE

Dear Dr. kawata,

Thank you for submitting your manuscript to PLOS ONE. After careful consideration, we feel that it has merit but does not fully meet PLOS ONE’s publication criteria as it currently stands. Therefore, we invite you to submit a revised version of the manuscript that addresses the points raised during the review process.

We would appreciate receiving your revised manuscript by Apr 09 2020 11:59PM. To enhance the reproducibility of your results, we recommend that if applicable you deposit your laboratory protocols in protocols.io, where a protocol can be assigned its own identifier (DOI) such that it can be cited independently in the future. For instructions see: http://journals.plos.org/plosone/s/submission-guidelines#loc-laboratory-protocols

We look forward to receiving your revised manuscript.

Kind regards,

Tatsuo Shimosawa, M.D., Ph.D.

Academic Editor

PLOS ONE

Journal Requirements:

2. Please include the total number of animals used in your study and the IACUC approval number related to your study. In the Methods section regarding the CKD rats with SHPT induced by 5/6 Nx please include the frequency of animal monitoring, including the specific criteria you used to monitor animal health.

3. In regard to Evocalcet, please provide more information in the Methods regarding product lot number.

4. We noticed you have some minor occurrence(s) of overlapping text with the following previous publication(s), which needs to be addressed:

https://doi.org/10.1371/journal.pone.0195316

In your revision ensure you cite all your sources (including your own works), and quote or rephrase any duplicated text outside the Methods section. Further consideration is dependent on these concerns being addressed.

5. Thank you for stating the following in the Competing Interests/Financial Disclosure* (delete as necessary) section:

We note that one or more of the authors are employed by a commercial company: Kyowa Kirin Co., Ltd..

Reviewers' comments:

Reviewer's Responses to Questions

**Comments to the Author**

1. Is the manuscript technically sound, and do the data support the conclusions?

Reviewer #1: Yes

Reviewer #2: Yes

2. Has the statistical analysis been performed appropriately and rigorously? 

Reviewer #1: Yes

Reviewer #2: Yes

3. Have the authors made all data underlying the findings in their manuscript fully available?

Reviewer #1: Yes

Reviewer #2: Yes

4. Is the manuscript presented in an intelligible fashion and written in standard English?

Reviewer #1: Yes

Reviewer #2: Yes

5. Review Comments to the Author

Reviewer #1: Congratulations on this excellent study. I think this was well conducted and well explained. I found a few problems with grammar, sentence structure and syntax that I have tried to correct below. Kindly take a look. There were a couple of clarifications that I requested on subject matter as noted below.

Line 42: Would change to: ‘…Since there is no treatment that can efficiently reverse vascular calcification, its prevention is important in improving prognosis of patients with SHPT’

Line 45: Change to : ‘Parathyroid hyperplasia is a characteristic feature of SHPT and its progression leads to hypersecretion of PTH and this in turn leads to altered mineral metabolism.’

Line 50: Add comma after therefore: ‘…therefore, been widely used…’

Line 52: Change to : ‘It has been reported that …’

Line 55: Change to : ‘It has also been suggested…’

Line 57: Change to: ‘However, cinacalcet has been associated with gastro-intestinal (GI) side-effects such as nausea and vomiting, …’

Line 61: Change to : ‘ …similar pharmacological profile to cinacalcet, it has lesser effect…’

Line 63: Change to: ‘…hyperplasia need further evaluation” or “ …need to be evaluated further.’

Line 104: Consider modifying- It is unclear what you are trying to say (what do you mean by established ?) – I think you could modify it to: “CKD rats with SHPT induced by adenine by the methods described above, were used.”

Line 272: Change to: ‘…that improves several issues with cinacalcet, including the high rate of GI side effects, …’

Line 281: Do you mean to say that the dosage of of Evocalcet (0.3mg/kg) that showed clear suppressive effect on serum PTH and Ca levels IN PREVIOUS STUDIES by single administration was used in this study? … If so, kindly add that phrase.

Line 288-291: I think studies (including reference 27 that you have quoted) show that cinacalcet and evocalcet decrease serum phosphorus levels only in dialysis patients, not in all CKD patients. In fact, due to the negation of phosphaturic effect of PTH by these calcimimetics, phosphorus levels actually INCREASE in CKD patients (see PMID:23023638, DOI: CliCa121015671576). What you found in your study, where phosphorus levels increased in CKD rats treated with evocalcet, is consistent with the above studies. I think you tried to explain this concept in lines 292 to 295 but would clarify lines 288-290 to reflect the above.

Line 294: Change ‘cancels’ to ‘negates’

Line 310: Change ‘As a result, evocalcet significantly…’ to ‘We found that, evocalcet significantly…’

Line 317 to 318: Change to: “These reports suggest that activation of CaR is important in the suppression of parathyroid cell proliferation.”

Finally, in table 2, Adenine induced CKD rats treated with evocalcet seemed to have had increased renal calcification, out of proportion to other organs, almost comparable to vehicle treated rats (though the degree of calcification seems to be slightly less). I am just curious as to what your thoughts are about this finding.

Reviewer #2: Authors clarified suppressive effects of Evocalcet on tissue calcification and parathyroid hyperplasia induced by kidney dysfunction.

1) In Line 226, authors used 2 doses of Evocalcet (0.03 or 0.3 mg/kg), although results from only 1 dose were shown in Figure 2 and Figure 3.

2) Did authors look at changes of parathyroid cell apoptosis, expression of p21 and CaR, and an oxyphil/chief cell ratio after Evocalcet administration?

3) What is the mechanism of cell size reduction after Evocalcet administration? Cellular structure, component of organella, and PTH content may be examined.

6. PLOS authors have the option to publish the peer review history of their article (what does this mean?). If published, this will include your full peer review and any attached files.

Reviewer #1: Yes: Elijah V. Kakani

Reviewer #2: Yes: Shozo Yano

---

## [Author Response · Author response to Decision Letter 0]

8 Apr 2020

Dear Academic Editor

Tatsuo Shimosawa, M.D., Ph.D.

Academic Editor

PLOS ONE

Title: Evocalcet prevents ectopic calcification and parathyroid hyperplasia in rats with secondary hyperparathyroidism 

Authors: Mariko Sakai, Shin Tokunaga, Mika Kawai, Miki Murai, Misaki Kobayashi, Tetsuya Kitayama, Satoshi Saeki, and myself.

We thank the referees for the carefully reviewing our manuscript and fruitful suggestions, especially for suggesting better terms and sentences concerning our manuscript. We found the comments of the reviewers most helpful and thus revised our manuscript according to the suggestions of the reviewers. We enclose revised versions of the manuscript, both with highlights of all changes and with unmarked changes. We also include a letter of our itemized responses to the Editor’s and Reviewer’s comments. We hope that this revised manuscript is now acceptable for publication in PLOS ONE.

About Funding and Competing Interests, 

Mitsubishi Tanabe Pharma Corporation provided evocalcet. All studies were performed and the cost of them were supported by Kyowa Kirin Co., Ltd. Mariko Sakai, Shin Tokunaga, Mika Kawai, Miki Murai, Misaki Kobayashi, Tetsuya Kitayama, Satoshi Saeki, and Takehisa Kawata are employees of Kyowa Kirin Co., Ltd. Kyowa Kirin Co., Ltd provided support in the form of salaries for authors, MS, ST, MK, MM, MK, TK, SS and TK, but did not have any additional role in the study design, data collection and analysis, decision to publish, or preparation of the manuscript. The specific roles of these authors are articulated in the ‘author contributions’ section.

Evocalcet is the product in development, and Kyowa Kirin Co., Ltd and Mitsubishi Tanabe Pharma Corporation have the ownership of the patents of evocalcet. 

These does not alter our adherence to PLOS ONE policies on sharing data and materials.

Thank you again for considering our manuscript. We are looking forward to hearing from you soon. 

Respectfully yours,

Takehisa Kawata, PhD

Nephrology Research Laboratories, Nephrology R&D Unit, R&D Division

Kyowa Kirin Co., Ltd

1188, Shimotogari, Nagaizumi-cho, Sunto-gun, Shizuoka, 411-8731, Japan

Phone: +81-90-2047-4856

Fax: +81-55-986-7430

e-mail: Takehisa.kawata.kk@kyowakirin.com

 

Journal Requirements:

Response; We checked the format to make sure it was correct for PLOSONE style requirements. If there are any problems, please let us know.

2. Please include the total number of animals used in your study and the IACUC approval number related to your study. In the Methods section regarding the CKD rats with SHPT induced by 5/6 Nx please include the frequency of animal monitoring, including the specific criteria you used to monitor animal health.

Response; Thanks for your helpful suggestions. We have noted animal numbers in each study in the method section (page 6, line 106, page 6, line 108, page 7, line 117, page 9, line 154). 

We have also noted IACUC number as follows (page 6, line 98-102). 

“protocol number APS 18J0188 for the single administration study, 17J0078 for the five-week administration study using CKD rats with SHPT induced by adenine, 14J0052 for the four-week administration study using CKD rats with SHPT induced by 5/6 Nx”

We revised the manuscript and described frequency of animal monitoring, including the specific criteria as follows (page 5, line 86-90).

“All experimental animals were monitored at least once per working day throughout the course of the study. Humane endpoints were defined as reduced physical activity level, weight loss, hunched posture, and other signs of distress. All rats reaching humane endpoints or in the single administration study were euthanized by carbon dioxide inhalation after the completion of studies.”

3. In regard to Evocalcet, please provide more information in the Methods regarding product lot number.

Response; Thanks for your helpful comment. We have noted the lot number of evocalcet in the method section (page 5, line73-75).

4. We noticed you have some minor occurrence(s) of overlapping text with the following previous publication(s), which needs to be addressed:

https://doi.org/10.1371/journal.pone.0195316

In your revision ensure you cite all your sources (including your own works), and quote or rephrase any duplicated text outside the Methods section. Further consideration is dependent on these concerns being addressed.

Response; I am very sorry that some of the expressions were duplicated from past our paper. We corrected the following points.

Page 3, line 50-54

 “Cinacalcet is the first approved calcimimetic agent in 2004, allosterically modulates the calcium receptor (CaR) on parathyroid gland cells and suppresses PTH secretion [4, 5]. Cinacalcet has been widely used clinically for more than 10 years in the world for management of SHPT in dialysis patients [6-11] and succeeded to reduce the number of parathyroidectomies [12].”

Page 4, line 59-62

“Although cinacalcet has excellent PTH lowering effect, it has problems of causing upper gastro-intestinal (GI) side effects such as nausea and vomiting at a certain rate [17]. These problems sometimes become an obstacle to the long-term use of cinacalcet or treatment with increased doses of cinacalcet [18, 19].”

5. Thank you for stating the following in the Competing Interests/Financial Disclosure* (delete as necessary) section:

We note that one or more of the authors are employed by a commercial company: Kyowa Kirin Co., Ltd..

Response; Sorry for not enough information. Kyowa Kirin Co., Ltd. provided authors’ salaries and research fundings. We add following sentences at Disclosures section as follows (page 21, line 359-370).

“About Funding and Competing Interests, Mitsubishi Tanabe Pharma Corporation provided evocalcet. All studies were performed and the cost of them were supported by Kyowa Kirin Co., Ltd. Mariko Sakai, Shin Tokunaga, Mika Kawai, Miki Murai, Misaki Kobayashi, Tetsuya Kitayama, Satoshi Saeki, and Takehisa Kawata are employees of Kyowa Kirin Co., Ltd. Kyowa Kirin Co., Ltd provided support in the form of salaries for authors, MS, ST, MK, MM, MK, TK, SS and TK, but did not have any additional role in the study design, data collection and analysis, decision to publish, or preparation of the manuscript. The specific roles of these authors are articulated in the ‘author contributions’ section. Evocalcet is the product in development, and Kyowa Kirin Co., Ltd and Mitsubishi Tanabe Pharma Corporation have the ownership of the patents of evocalcet. These does not alter our adherence to PLOS ONE policies on sharing data and materials.”

Response; There are no clinical data, so we feel that this comment is not relevant to our manuscript.

ーーーーーーーーーーーーーーーーーーーーーーーーーーーーーーーーーーーーーーーーーーーーーーーーーーーーーーーーーーーーーー

Reviewer #1: Congratulations on this excellent study. I think this was well conducted and well explained. I found a few problems with grammar, sentence structure and syntax that I have tried to correct below. Kindly take a look. There were a couple of clarifications that I requested on subject matter as noted below.

Line 42: Would change to: ‘…Since there is no treatment that can efficiently reverse vascular calcification, its prevention is important in improving prognosis of patients with SHPT’

Response; Thanks for your helpful comment. We have revised the sentence as you pointed out (line 43-45, page 3). 

Line 45: Change to : ‘Parathyroid hyperplasia is a characteristic feature of SHPT and its progression leads to hypersecretion of PTH and this in turn leads to altered mineral metabolism.’

Response; Thanks for your helpful comment. We have revised the sentence as you pointed out (line 46-48, page 3).

Line 50: Add comma after therefore: ‘…therefore, been widely used…’

Response; Thank you very much for your helpful comment. The editor also instructed us to rewrite this part because it is similar to our previous papers, so we completely revised it as follows (line 50-54, page 3). 

“Cinacalcet is the first approved calcimimetic agent in 2004, allosterically modulates the calcium receptor (CaR) on parathyroid gland cells and suppresses PTH secretion [4, 5]. Cinacalcet has been widely used clinically for more than 10 years in the world for management of SHPT in dialysis patients [6-11] and succeeded to reduce the number of parathyroidectomies [12].”

Line 52: Change to : ‘It has been reported that …’

Response; Thanks for your helpful comment. We have revised the sentence as you pointed out (line 54, page 4).

Line 55: Change to : ‘It has also been suggested…’

Response; Thanks for your helpful comment. We have revised the sentence as you pointed out (line 57, page 4).

Line 57: Change to: ‘However, cinacalcet has been associated with gastro-intestinal (GI) side-effects such as nausea and vomiting, …’

Response; Thank you very much for your helpful comment. The editor also instructed us to rewrite this part because it is similar to our previous papers, so we completely revised it as follows (line 59-60, page 4). 

“Although cinacalcet has excellent PTH lowering effect, it has problems of causing upper gastro-intestinal (GI) side effects such as nausea and vomiting at a certain rate [17]. These problems sometimes become an obstacle to the long-term use of cinacalcet or treatment with increased doses of cinacalcet [18, 19].”

Line 61: Change to : ‘ …similar pharmacological profile to cinacalcet, it has lesser effect…’

Response; Thanks for your helpful comment. We have revised the sentence as you pointed out (line 64, page 4).

Line 63: Change to: ‘…hyperplasia need further evaluation” or “ …need to be evaluated further.’

Response; Thanks for your helpful comment. We have revised the sentence as you pointed out (line 64-65, page 4).

Line 104: Consider modifying- It is unclear what you are trying to say (what do you mean by established ?) – I think you could modify it to: “CKD rats with SHPT induced by adenine by the methods described above, were used.”

Response; Thanks for your helpful comment. We have revised the sentence as you pointed out (line 116, page 7).

Line 272: Change to: ‘…that improves several issues with cinacalcet, including the high rate of GI side effects, …’

Response; Thanks for your helpful comment. We have revised the sentence as you pointed out (line 282-283, page 17).

Line 281: Do you mean to say that the dosage of of Evocalcet (0.3mg/kg) that showed clear suppressive effect on serum PTH and Ca levels IN PREVIOUS STUDIES by single administration was used in this study? … If so, kindly add that phrase. 

Response; Thanks for your helpful comment. We have revised the sentence as you pointed out and added reference (line 291-292, page 18).

Line 288-291: I think studies (including reference 27 that you have quoted) show that cinacalcet and evocalcet decrease serum phosphorus levels only in dialysis patients, not in all CKD patients. In fact, due to the negation of phosphaturic effect of PTH by these calcimimetics, phosphorus levels actually INCREASE in CKD patients (see PMID:23023638, DOI: CliCa121015671576). What you found in your study, where phosphorus levels increased in CKD rats treated with evocalcet, is consistent with the above studies. I think you tried to explain this concept in lines 292 to 295 but would clarify lines 288-290 to reflect the above.　 

Response; Thank you for your valuable suggestion. We cited the original reference from the reference you pointed out and revised the sentence as follows (line 298-302, page 18). 

“Although evocalcet tends to decrease the serum phosphate levels in patients with SHPT, we could not observe the reduction in serum IP levels in CKD rats with SHPT induced by adenine. The discrepancy is possibly due to the remnant kidney function of this animal model of CKD. In fact, due to the negation of phosphaturic effect of PTH by these calcimimetics, phosphorus levels actually increase in CKD patients [28].”

Line 294: Change ‘cancels’ to ‘negates’

Response; Thanks for your helpful comment. We have revised the sentence as you pointed out (line, page).

Line 310: Change ‘As a result, evocalcet significantly…’ to ‘We found that, evocalcet significantly…’

Response; Thanks for your helpful comment. We have revised the sentence as you pointed out (line, page).

Line 317 to 318: Change to: “These reports suggest that activation of CaR is important in the suppression of parathyroid cell proliferation.”

Response; Thanks for your helpful comment. We have revised the sentence as you pointed out (line 304, page 18).

Finally, in table 2, Adenine induced CKD rats treated with evocalcet seemed to have had increased renal calcification, out of proportion to other organs, almost comparable to vehicle treated rats (though the degree of calcification seems to be slightly less). I am just curious as to what your thoughts are about this finding.

Response; Thank you very much for your helpful comments. As you say, calcification of the kidney seemed to be not decreased enough compared to other organs in the evocalcet group. The reason is not clear, so it is only a hypothesis, but we think the reason of it as follows. The renal calcification is induced not only blood vessels but also renal tubules by lot excretion of calcium and phosphorus in adenine rats. Evocalcet reduced serum PTH and promotes calcium excretion. Therefore, even if serum calcium was lowered, the calcium concentration in the renal tubules might have remained high, and it is considered that the calcification did not improve sufficiently. This calcimimetic’s effect on urinary Ca excretion become a concern when renal function is present, but we do not believe this problem is a major clinical problem in dialysis patients.

Reviewer #2: Authors clarified suppressive effects of Evocalcet on tissue calcification and parathyroid hyperplasia induced by kidney dysfunction.

1) In Line 226, authors used 2 doses of Evocalcet (0.03 or 0.3 mg/kg), although results from only 1 dose were shown in Figure 2 and Figure 3.

Response; Thanks for your helpful comment. It’s just our mistake and sorry for about that. We have revised the sentence as you pointed out (line 237, page 14).

2) Did authors look at changes of parathyroid cell apoptosis, expression of p21 and CaR, and an oxyphil/chief cell ratio after Evocalcet administration?

Response; Thank you for pointing it out. In this study, we did not observe the changes of p21 and CaR expression, apoptosis, and an oxifil/chief cell ratio because we could not establish an excellent system, even though we studied it.

3) What is the mechanism of cell size reduction after Evocalcet administration? Cellular structure, component of organella, and PTH content may be examined.

Response; Thank you for a very useful question. The size of the parathyroid gland depends on the number and size of the cells. Although this is not a thorough study, we have an impression that the size of glands decreases in a short period of time, even though the number of cells does not change, when calcimimetics is administered to CRF rats with enlarged glands caused by SHPT. Therefore, the mechanism by which cells decrease in size is largely due to the decrease in the amount of endoplasmic reticulum containing PTH.

---

## [Decision Letter · Decision Letter 1]

15 Apr 2020

Evocalcet prevents ectopic calcification and parathyroid hyperplasia in rats with secondary hyperparathyroidism.

PONE-D-20-01532R1

Dear Dr. kawata,

We are pleased to inform you that your manuscript has been judged scientifically suitable for publication and will be formally accepted for publication once it complies with all outstanding technical requirements.

With kind regards,

Tatsuo Shimosawa, M.D., Ph.D.

Academic Editor

PLOS ONE

Additional Editor Comments (optional):

Reviewers' comments:

Reviewer's Responses to Questions

**Comments to the Author**

1. If the authors have adequately addressed your comments raised in a previous round of review and you feel that this manuscript is now acceptable for publication, you may indicate that here to bypass the “Comments to the Author” section, enter your conflict of interest statement in the “Confidential to Editor” section, and submit your "Accept" recommendation.

Reviewer #1: All comments have been addressed

Reviewer #2: All comments have been addressed

2. Is the manuscript technically sound, and do the data support the conclusions?

Reviewer #1: (No Response)

Reviewer #2: Yes

3. Has the statistical analysis been performed appropriately and rigorously? 

Reviewer #1: (No Response)

Reviewer #2: Yes

4. Have the authors made all data underlying the findings in their manuscript fully available?

Reviewer #1: (No Response)

Reviewer #2: Yes

5. Is the manuscript presented in an intelligible fashion and written in standard English?

Reviewer #1: (No Response)

Reviewer #2: Yes

6. Review Comments to the Author

Reviewer #1: (No Response)

Reviewer #2: Authors have made suitable comments and corrections raised by reviewers and editors. Thus, I do not think further concerns are present.

7. PLOS authors have the option to publish the peer review history of their article (what does this mean?). If published, this will include your full peer review and any attached files.

Reviewer #1: No

Reviewer #2: Yes: Shozo Yano

---

## [Editor Report · Acceptance letter]

17 Apr 2020

PONE-D-20-01532R1 

Evocalcet prevents ectopic calcification and parathyroid hyperplasia in rats with secondary hyperparathyroidism. 

Dear Dr. kawata:

I am pleased to inform you that your manuscript has been deemed suitable for publication in PLOS ONE. Congratulations! Your manuscript is now with our production department. 

With kind regards,

on behalf of

Prof. Tatsuo Shimosawa 

Academic Editor

PLOS ONE